# Sustainable Extraction of Prospective Cosmetic Ingredients from Colombian Marine Macroalgae Using Natural Deep Eutectic Solvents

**DOI:** 10.3390/md23060239

**Published:** 2025-05-30

**Authors:** Verónica María Tamayo-Rincón, Jhonny Colorado-Ríos, Didier Johan Alvarez-Bustamante, Vanessa Urrea-Victoria, Diana Margarita Márquez-Fernández, Constain H. Salamanca, Stefano Dall’Acqua, Leonardo Castellanos-Hernandez, Alejandro Martínez-Martínez

**Affiliations:** 1Grupo Productos Naturales Marinos, Facultad de Ciencias Farmacéuticas y Alimentarias, Universidad de Antioquia, Calle 70 No. 52-21, Medellín 050010, Colombia; veronica.tamayo@udea.edu.co (V.M.T.-R.); jhonny.colorado@udea.edu.co (J.C.-R.); djohan.alvarez@udea.edu.co (D.J.A.-B.); diana.marquez@udea.edu.co (D.M.M.-F.); 2Unidad de Biología Celular y Molecular, Corporación para Investigaciones Biológicas (CIB), Universidad de Antioquia, Calle 70 No. 52-21, Medellín 050010, Colombia; 3Departamento de Química, Facultad de Ciencias, Universidad Nacional de Colombia, Sede Bogotá, Av. Carrera 30 # 45-03, Bogotá 111321, Colombia; vurreav@unal.edu.co (V.U.-V.); lcastellanosh@unal.edu.co (L.C.-H.); 4Grupo de Investigación Biopolimer, Departamento de Farmacia, Facultad de Ciencias Farmacéuticas y Alimentarias, Universidad de Antioquia, Calle 70 No. 52-21, Medellín 050010, Colombia; chsm70@gmail.com; 5Department of Pharmaceutical and Pharmacological Sciences, University of Padova, Via Marzolo 5, 35121 Padova, Italy; stefano.dallacqua@unipd.it

**Keywords:** NADES, macroalgae, marine natural products, natural cosmetics, green chemistry

## Abstract

This study presents the results obtained from extracting and quantifying cosmetically valuable metabolites such as phenolic compounds and mycosporine-like amino acids (MAAs) from 12 samples of marine macroalgae collected in the Colombian Caribbean Sea. Natural deep eutectic solvents (NADESs) were prepared, physicochemically tested (viscosity, surface tension, pH, and conductivity), and then compared with water as the reference solvent to quantify phenolic compounds using the Folin–Ciocalteau test. With a simple extraction assay with water and ultrasound followed by ultraviolet spectral scanning the presence of MAAs was easily determined in several of the analysed samples, and then they were identified by HPLC-DAD. Hydrochloric acid solution at 5% extracted a higher content of phenolic compounds than NADES and water. The NADES that showed the highest phenolic compound extraction yield was a mixture of betaine, glucose, and water with 1:1:5 molar ratio. *Sargassum* cf. *ramifolium* and *Sargassum fluitans* showed the highest contents of phenolic compounds extracted with NADES, with 29.2 and 21.9 mg GAE/g DW, respectively. The results show that NADESs are an interesting alternative for the more efficient extraction of cosmetically valuable compounds such as phenolic compounds and mycosporine-type amino acids from marine macroalgae.

## 1. Introduction

Marine macroalgae are known as sources of substances of nutritional, cosmetic, pharmaceutical, and industrial interest. These substances have been extracted for several decades using organic solvents including alcohols, halogenated compounds, and acetone, among others. Generally, organic solvents generate waste that contaminates the air, soil, and water sources for human use, which has led various researchers in natural product chemistry to develop and test alternatives that are less hazardous to the environment and, therefore, to the health of living beings. These new alternatives include the natural deep eutectic solvents (NADESs). These solvents are easily prepared by combining naturally occurring substances in different proportions. The name “eutectic” derives from the physicochemical fact that combining a solid that acts as a hydrogen donor with another that acts as a hydrogen acceptor in certain equimolar quantities results in a mixture with a lower melting point than the pure compounds due to the formation of hydrogen bonds between the two substances. The results are eutectic mixtures that are viscous liquids at low temperatures. The viscosity of these NADES liquids can be reduced by adding water in certain proportions, and they are thus used for the extraction of substances from natural sources [1,2]. NADESs have become a proposal in Green Chemistry to partially or completely replace conventional organic solvents, as they do not generate waste as toxic as the latter and may provide higher yields in the extraction of various substances, such as phenolic compounds.

Extraction with NADESs has been used for about 15 years, and many scientific reports have been found in studies with terrestrial plants, but very few have been found with marine organisms in general [3,4,5]. Examples of some processes including NADESs for the cosmetic industry include the extraction of phenolic acids and flavonoids from terrestrial plants like the flowers of *Calendula officinalis* with mixtures of fructose/glycerine, fructose/betaine, and fructose/sorbitol, among others. Each of them exhibited an extraction efficiency towards phenolic acids equal to or higher than the best conventional solvents such as methanol, ethanol, acetone, hexane, ethyl acetate, chloroform, dichloromethane, etc. [6]. One example is the extraction of phenylethanes and phenylpropanoids from *Rhodiola rosea* L., which used an L-lactic acid/fructose-based NADES that was proposed as a viable alternative to 40% aqueous ethanol for the extraction of salidroside, tyrosol, rosavin, rosin, and cinnamyl alcohol [7].

In the particular case of marine macroalgae, a review of the existing literature found a few reports related to the use of NADESs for the extraction of secondary metabolites, especially focused on the extraction of phlorotannins and trace metals from *Fucus vesiculosus* [8,9], fatty acids from *Spirulina* [10], phycoerythrin from *Porphyra yezoensis* [11], α-chitin from crab *Polybius henslowii* [12], and the extraction of phlorotannins from the brown macroalga *Ascophyllum nodosum* L. [13]. The extraction efficiency of polyphenolic compounds was evaluated using ten types of NADES based on choline chloride, lactic acid, betaine, and glucose in various molar proportions [6]. Another study focused on red macroalgae, specifically *Gelidium corneum* (Hudson) J.V. Lamouroux and *Sargassum muticum*, to obtain antioxidant-enriched extracts using NADESs to introduce into cosmetic formulations. That study highlighted the role of NADESs to address the lack of information concerning their use in the extraction of seaweed phenolic compounds and their application in a topical formulation for dermatological use. Extraction yields with NADESs surpassed those of conventional solvents like water and ethanol (70:30, *v*/*v*) in the two extracted seaweeds. Furthermore, *S. muticum* extracts revealed higher antioxidant capacity strongly related to their high phenolic content, proving to be suitable for further dermatological applications with antioxidant properties.

Another example is the study in which the authors extracted enzyme-inhibiting substances from *Hypnea flagelliformis*. They used mixtures of choline chloride and lactic acid to obtain different NADESs. When compared with the extractions using 80% ethanol and 80% methanol, the yields obtained with the NADESs choline chloride/lactic acid 1:2, 80% ethanol, 80% methanol, and water were 92.24, 88.33, 86.76, and 54.33 mg GAE/100 g DW, respectively [14]. It is important to note that the use of choline derivatives is not recommended, as it is considered a prohibited substance for cosmetic purposes (European Commission). This study is a preliminary contribution to establish which physicochemical properties could be correlated with the extraction of compounds from macroalgae despite the lack of comprehensive investigation for physicochemical properties for non-choline-based NADESs.

Colombia is in the planet’s tropical zone, with coastlines on both the Atlantic and Pacific Oceans, and research is underway mainly on the macroalgae found in the Caribbean Sea. Macroalgae and most Colombian biodiversity are chemically unexplored. The environmental conditions of the Caribbean are highly favourable for the development of a unique biodiversity of red and brown macroalgae, groups that are highly heterogeneous in terms of form and growth. They depend on ecosystem environments such as rocky coasts, reefs, mangroves, grasslands, and sandbanks, with numerous interactions between different organisms. This has led macroalgae to develop chemical defence mechanisms that result in a significant production of bioactive metabolites to survive in this particularly competitive environment [15]. This study reports the results of extraction processes for phenolic compounds in twelve unexplored samples of Colombian marine macroalgae and reports a simple test for the detection of mycosporine-like amino acids using water and NADESs.

## 2. Results and Discussion

### 2.1. Samples

This study focuses on developing simple eco-friendly processes for several species to evaluate the potential use for cosmetic applications of twelve marine macroalgae collected from northeast Colombian coastal areas. These processes may help create products with relevant properties. The phenolic compounds were extracted from the collected samples under ultrasound-assisted extraction with NADESs and water as a reference solvent. The selected macroalgae specimens belong to the Phaeophyta (brown macroalgae) and Rhodophyta (red macroalgae) phyla, as outlined in Table 1. These macroalgae contain compounds that are claimed to have cosmetic applications, including polysaccharides, minerals, lipids, proteins, phenolic compounds, terpenes, halogen compounds, carotenoids, vitamins, and sulphur and nitrogen derivatives. In this study, the extraction focused specifically on phenolic compounds (PCs) and mycosporine-like amino acids (MAAs), which are recognized for their antioxidative and photoprotective properties, respectively [16]. Low concentrations of Cd (up to 0.4 μg/kg) and Hg (up to 3.5 μg/kg), and the absence of Pb (<LOQ) were observed from a random selection of samples collected in the same zones of this study and tested by atomic absorption spectroscopy.

### 2.2. Preliminary Chemical Screening

Preliminary phytochemical assays were conducted based on solid–liquid extractions, qualitative test reactions, and thin-layer chromatography to check representative compounds such as low-polarity compounds, alkaloids, polysaccharides, MAAs, and phenolic compounds. All macroalgae samples extracted with ethanol, water, and 5% HCl solution displayed positive results for polysaccharides; the alkaloids tests (precipitation with Wagner’s reagent) were negative for all macroalgae extracted with 5% HCl solution; phenolic compound reactions with 1% ferric chloride solution displayed positive results for all of the macroalgae samples extracted with 70% ethanol and with water; and MAAs were detected by ultraviolet spectroscopy in the aqueous extracts from *C. nitens*, *G. lemaneiformis,* and *P. capillacea*.

### 2.3. Preparation and Analysis of NADESs

NADESs are green alternatives to the use of traditional organic solvents due to their low toxicity, biodegradability, and ability to solubilize bioactive compounds. NADESs are formed by mixing two or more natural compounds, which could be solids or liquids. The preparation of NADESs was achieved by mixing hydrogen bond acceptors (HBAs) with hydrogen bond donors (HBDs) with molar relations based on the literature [17] and avoiding the use of choline chloride due to its restriction from being used in cosmetics formulations despite being broadly used for the preparation of NADESs. Preliminary experiments led us to select three transparent colourless liquids as the NADES systems (Table 2). However, it is well known that the physicochemical properties of eutectic mixtures are dependent on different factors including molar ratio, water content, type of HBD, size of chain, temperature, etc. [4,17,18]. It is also important to mention that the selection for physicochemical properties evaluated here is empirical. We consider that these would be valuable for future investigation about the full characterization of physicochemical properties or the control quality and stability of extracts based on NADESs, which is not the focus of this study; therefore, no deep discussion will be presented. Given our specific NADES combinations in Table 2, it is very important to establish which properties could be correlated and how strong are their relationships with the extraction of compounds from macroalgae despite the lack of comprehensive investigation for physicochemical properties for non-choline-based NADESs.

#### 2.3.1. Flow Profile of NADESs

Viscosity is one of the most important parameters that characterize eutectic mixtures, as it provides a good basis for understanding their interactions at the molecular level in the liquid phase by affecting the solubility behaviour of macroalgae compounds in the different NADESs because of its influence on the rate of diffusion of metabolites from the solid sample to the liquid phase, thus limiting the time of extraction.

At the viscosity level, the NADESs containing water (BGlcW115 and FGlcW115) are, in principle, much less viscous than that lacking water (UGly13). The existence of an inherently very viscous component such as glycerol in the mixture UGly13, with negligible water content, perfectly justifies its position as the most viscous system being used in this study (Figure 1). At the same time, the higher water contents of BGluW115 and FGlcW115 may explain their lower viscosities. Figure 1 shows that the NADESs prepared have a Newtonian behaviour due to their constant viscosities over time (shear rate), having linear behaviour.

#### 2.3.2. Surface Tension of NADESs

Along with viscosity, surface tension has been one of most-studied physical properties in eutectic mixture fields. The experimental values of surface tensions are presented in Table 3, which also includes the conductivity and pH results of the same mixtures. Analysing the surface tensions of the investigated NADESs, listed in Table 3, it is possible to conclude that BGlcW115 displays higher cohesive energies than the other NADESs. In fact, this would be expected for all NADESs in this study, since all the components of these mixtures contain one or more polar groups that lead to the formation of a vast network of hydrogen bonds, which attract and hold molecules close to each other, consequently, increasing the energy that is sent to hold the surface area. The surface tension values presented in Table 3 are within the same range of values found for the other hydrophilic systems [19].

#### 2.3.3. pH of NADESs

In Table 3, the pH values of the NADESs varied notably. The FGlcW115 mixture exhibited a neutral pH of 7.38, indicating a balanced environment that is typically conducive to the solubilization of a wide range of compounds, including phenolic compounds. In contrast, BGlcW115 had a higher pH of 8.44, suggesting a more alkaline environment. This increase in pH could have enhanced the solubility of certain phenolic compounds, particularly those with acidic functional groups, due to the deprotonation of hydroxyl groups, thus facilitating their extraction. UGly13 presented the highest pH at 9.30, which may indicate a stronger alkaline character. This high-pH environment could promote the extraction of specific polar compounds, potentially altering their chemical state and enhancing their solubility. However, the elevated pH may also pose risks, as it could lead to the degradation of sensitive biomolecules during the extraction process.

#### 2.3.4. Conductivity of NADESs

The conductivity measurements reflect the ionic strength and overall electrochemical environment of the NADESs. FGlcW115 showed the highest conductivity at 2.2 µS/cm, which aligns with its neutral pH. This increased conductivity indicates a higher concentration of ions in solution, which can enhance the solubilization capacity for polar compounds. Conversely, BGlcW115 exhibited lower conductivity at 0.9 µS/cm. This reduced conductivity may be attributed to the unique interactions of betaine with water and glucose, potentially leading to a more stabilized environment for solubilizing compounds without a significant ionic presence. UGly13 had a conductivity of 1.4 µS/cm. This suggests a moderate ionic environment, potentially allowing for effective extraction while avoiding excessive ionic interactions that could hinder the solubilization of certain compounds.

### 2.4. Ultrasound-Assisted Extraction of Phenolic Compounds with the NADESs

The extraction of bioactive compounds using NADESs is influenced by the high viscosity that can limit their use as extractants compared with conventional solvents such as water or ethanol due to the lower mass-transport efficiency. To overcome this major drawback, increases in the temperature and the water content were applied to the mixtures FGlcW115 and BGlcW115 in order to decrease the viscosity, thus intensifying the mass transfer through solid–liquid contact between raw material and NADES [20]. Considering that some natural compounds are sensitive to elevated temperatures [21], the extraction efficiency was promoted just by applying a temperature increase up to 60 °C. The crushed raw material was used in a solid–liquid (S/L) ratio of 1:15 (0.100 g in 1.5 mL), according to reports of S/L at 1:10 or 1:20 suggested as adequate to extract phenolic compounds [22,23]. The other extraction parameters, such as the extraction temperature of 60 °C and the ultrasound-assisted extraction time of 60 min, were held constant for all NADESs (Table 2). To obtain information about the influence of NADES type on extraction yield, water, as conventional solvent, was used as a contrastive solvent. The efficiency of ultrasound-assisted extraction (37 kHz for 60 min at 60 °C) was evaluated according to the total phenolic content (TPC) of the extracts expressed as mg GAE/g DW. BGlcW115 allowed us to extract the higher content of target compounds in most samples, and all NADESs showed higher extraction efficiencies than water (Figure 2, Table 4 and Appendix A).

The highest yields were observed for *Sargassum* spp. samples, with TPC values ranging from 6.86 to 29.2 mg GAE/g. *S.* cf *ramifolium* showed the highest TPC values, which were 12.8- and 6.53-fold higher than the lowest values for *G. lemaneiformis* and *Hypnea* sp. 2 (*p* < 0.05), respectively. The report about a sample from *Hypnea spinella* showed 2.019 ± 0.034 mg GAE/g DW, which is lower than our results for *Hypnea* spp. samples [24]. Moreover, TPC values for *S. fluitans* were the second highest and ranged from 15.1 to 21.9 mg GAE/g. NADES extracts from *Dictiota* spp. samples also showed high TPC values from 7.26 to 20.9 mg GAE/g.

Phenolics in seaweeds are synthesized via numerous metabolic pathways, contributing to the difference in existing forms, types, and content of phenolics [25]. Even the differences in the free or bound phenolics could be attributed to harvesting location and period, genetic factors, and extraction method [26,27], which explain discrepancies between previous data and our results that may be attributed to different extraction methods, harvest locations, and harvest periods affecting the accumulation of phenolic compounds [28]. Phenolic compounds are some of the bioactive compounds produced in seaweeds that have gained significant attention for skin care and are considered safe, with negligible cytotoxicity and many beneficial effects on humans. They vary quantitatively and qualitatively for each specimen of red, brown, or green seaweeds, with a wide range of studies and new developments in the pharmaceutical area and in other areas, where the predominant bioactivity of all is the antioxidative activity. To be more ecological and intuitive to perform, with better quality, purity, and quantity of the phenolic compounds extracted, extraction methods still need to be developed because seaweed phenolics can be key players in the future in different areas [29]. In this study, TPCs in brown seaweeds were significantly higher than those in other seaweeds, mostly when FGlcW115 and BGlcW115 were applied, which could be associated with the phenolic acids, bromophenols, flavonoids, and phlorotannins that are found in seaweeds, especially brown seaweeds [29].

UGly13 had the higher viscosity and the lower extraction efficiency, which is expected due to physicochemical principles, so between FGlcW115 and BGlcW115, the highest efficiency was directly correlated with the highest viscosity of BGlcW115. The reasons for those results were not only the effect of solvent viscosity but, more importantly, the effect of hydrogen bonds or ionic intermolecular forces formed between the NADESs and the target components. Therefore, the number of hydrogen bonds changed due to the water as well as the slight alkalinity added by betaine, which could produce partial ionization of phenolic compounds, thus contributing to the breakage of the samples’ cell walls and the dissolution of target components. Despite the differences in the phenolic compounds present in the 12 seaweeds under study, it can be concluded that the presence of betaine and water facilitated the extraction of phenolic compounds. Accordingly, the combination of BGlcW115 was chosen as the most promising for further research. Hydrochloric acid extraction showed a significantly better extraction than phenolic compounds in contrast to water (Figure 3), so it would be due to hydrolysis of fixed phenolic compounds that may not be extracted with water nor NADESs.

Although there are several important factors that must be taken into account, such as the constituents and the molar ratio of the NADES mixture, temperature, and water content, which are linked to viscosity, pH, polarity, and the surface tension of the NADES [30], the varying phenolic contents across the different seaweed species stem from multiple factors, such as species variations, plant growth stage, size, light exposure, etc. [31]. The largest proportion of phenolic compounds present in red seaweeds are of bromophenols, flavonoids, phenolics acids, phenolic terpenoids, and mycosporine-like amino acids [29]; phlorotannins are the major polyphenolic class (oligomers of phloroglucinol), found only in the marine brown seaweeds, exerting functions as primary and secondary metabolites [32]; and phenolic terpenoids have been detected and characterized in brown and red seaweeds [33]. Many of these phenolics are often bound with cellulose, pectin, and polysaccharides, making them difficult to hydrolyse [34], and they can interact with polysaccharides, with either synergistic or antagonistic effects [35]. Combinations with polysaccharide potentially alter their antioxidant activity by impeding scavenging of free radicals, or they can boost their antioxidant capacity in the case of the Folin reaction for impeding complex formation of phenolic compounds with metals (Mo/Bs) of the reagent solution. The structural compatibility and the relative ratios of polysaccharide and phenolic compounds, depending on the species, could govern their interactions and the displayed results differently. Phenolic compounds such as gallic acid, catechin, and phloroglucinol, which are commonly found in brown seaweed, contain benzene ring(s) and multiple -OH groups that can non-covalently interact with the sulphate and hydroxyl groups on fucoidan. This could potentially lead to entrapment of phenolic compounds or the formation of a carbohydrate–phenolic complex in brown as in red macroalgae, decreasing the ability of phenolic compounds to scavenge free radicals [35]. All the types of algae tested in this study were submitted to conditions for the extraction of free phenolics without having to account for the requirement to put together the most commonly used organic solvents for extracting free phenolics from seaweeds (particularly methanol, ethanol, acetone, their aqueous mixtures, and ethyl acetate) [34]. The extractability achieved using BGlcW115 was higher for the brown in comparison with the results for the red macroalgae, suggesting the necessity to continue the optimization method species by species. The fact that the chemical compositions of soluble and insoluble phenolic compounds in macroalgae determine a good or regular extraction for an established NADES led us to compare a preliminary comparison that was executed with an acid hydrolysis for extracting bound phenolics [36], even considering that alkaline hydrolysis was demonstrated to be more efficient for releasing bound phenolics than acid hydrolysis [34]. The TPCs of extracts released by acid hydrolysis were 1.64-, 14.85-, 3.62-, 9.19-, 3.39-, and 2.80-fold higher than those released by BGlcW115 for red macroalgae *C. nitens*, *G. lemaneiformis*, *Hypnea* sp. 1, *Hypnea* sp. 2, *P. capillacea,* and *S. filiformis*, respectively (*p* < 0.05). Acid hydrolysis was not enough to display a significant extraction of bounded phenolics in brown macroalgae; therefore, it needs to be compared with alkaline extraction as suggested previously for brown seaweed [26,35] in order to establish reference standards for the performance of NADESs, which are deemed to be potential extraction media because of their low toxicity and lack of adverse effects on the environment.

### 2.5. Pre-Scaling of Phenolic Compound Extraction

The initial screening of the selected NADESs enabled us to continue with the optimization of the process, aiming for the most suited conditions to increase the production of phenolic compounds based on BGlcW115 for the most promising seaweed, *Sargassum fluitans*, by its content of phenolic compounds and its abundance in the Colombian sea compared with other *Sargassum* species. The extraction efficiency is typically affected by several experimental conditions such as water content, molar ratio between NADESs’ components, and the conditions of extraction method, including temperature for conventional extraction and frequency when ultrasound-assisted extraction is used. Reported seaweed extractions usually employ high quantities of organic solvents (1:10, 1:20, 1:30, or as high as 1:100 g/mL) [37,38,39] followed by solvent-removal processes. In this study, the optimization process started with the following extraction conditions: extraction times of 1, 2, 6, 24, and 30 h; S/L ratio 1:10; and mechanical agitation at 60 °C without solvent removal. Firstly, the influence of the source of the NADESs’ components was studied. NADESs were prepared with raw materials from different suppliers for betaine and glucose and the same source of water at a molar ratio of 1:1:5, as previously tested. Figure 4A shows the time-dependent increasing of TPC from 1.0 g of dry weighted *S. fluitans* and extracted at 60 rpm and 60 °C with two different preparations of 10 mL of BGlcW115 during 30 h (BGlcW115-1 and BGlcW115-2). Both NADESs showed statistically different impacts on the total phenolic compounds (*p* < 0.05), except at the time of 30 h. At this time, at the point of maximal PC dissolution, it was possible to have the same results with NADESs prepared with raw materials from different suppliers. In BGlcW115-1, the maximum concentration was 5.23 mg GAE/g DW after 24 h of extraction time, and it remained stable until 30 h.

To evaluate the extraction efficiency, the S/L ratios were tested at 1:10, 1:20, and 1:50 to screen the performance and to preselect the best extraction conditions. Extractions were then carried out using 10, 20, and 50 mL of NADESs for 1.0 g of dry weighted macroalgae, with all the other conditions fixed, and monitoring the TPC at 6, 24, 30, 48, and 54 h in order to find the suitable S/L ratio and the extraction time for TPC. Since the solvent used for extraction will not be separated from the extract, the most concentrated extracts, those with the higher S/L ratios, will be favoured even if TPC values are lower regarding the initial mass of seaweed. Extraction data of this parameter are presented in Figure 4B and show a gradual increase in TPC due to the increase in the extraction time. The differences between S/L ratios 1:10, 1:20, and 1:50 are relatively small but significant (*p* < 0.05). This suggests that the extraction of phenolic compounds at S/L ratio of 1:10 reaches a superior level of extraction through the tested conditions. At 30 h of extraction, S/L ratios of 1:10 and 1:20 show remarkably similar percentages of phenolic compounds. This indicates that for short extraction times (up to 30 h), the S/L ratio is a determining factor in the extraction efficiency of phenolic compounds, suggesting that setting the volume of the solvent is necessary to obtain good returns within this time range. As is observed in Figure 4B, the NADES extraction yields oscillate in a very close range around 10 mg GAE/g DW, thus indicating there are low variations, and these were verified statistically (*p* > 0.9999). Accordingly, the extraction process has a high potential to be robust, with less efficiency than the ultrasound-assisted extraction. These conditions establish the experimental basis to develop the extraction method validation.

### 2.6. UV Preliminary Scanning of Mycosporine-like Amino Acids (MAAs)

In order to rapidly test mycosporine-like amino acids (MAAs), extracts were prepared with water and FGlcW115, BGlcW115, and UGly13 and displayed an absorption maximum between 324 and 326 nm, which are the characteristic absorption maxima for MAA when the samples are scanned between 268 and 362 nm, depending on their molecular structure. *Pterocladiella capillacea,* water, ethanol, BGlcW115, and UGly13 extracts displayed an absorption maximum between 320 and 330 nm; this absorption is characteristic for MAAs as reported [40]. 

### 2.7. Identification of MAAs by UHPLC-DAD in Red Macroalgae

The variability in the content of MAAs among macroalgal species is also influenced by the time of year, as indicated by the study of Urrea-Victoria et al. [41]. During different periods of the year, MAA concentrations in species collected from the Colombian Caribbean showed significant variations. For example, *Ceramium rubrum* exhibited 2.08 mg/g DW of shinorine, while *Gracilaria* sp. presented concentrations of palythine up to 56 mg/g DW, asterine up to 0.39 mg/g DW, and Porphyra-334 up to 8.44 mg/g DW. These differences reflect how environmental variations affect MAA production in macroalgae. These results underscore the importance of *Gracilariales* and *Ceramiales* species in MAA production, particularly for sunscreen products development.

The results from the identification of MAAs, as shown in the heatmap (Figure 5), provide a detailed view of their abundance based on the calculated areas. Notably, each species exhibited the presence of at least two out of the four MAAs: shinorine, palythine-330, asterine-330, and Porphyra-334. The results from MAA detection in macroalgae samples highlight the performance of various extraction methods.

Using water extraction (with the mobile phase in UHPLC-DAD analysis), all species exhibited detectable levels of shinorine, palythine-330, asterine-330, and Porphyra-334. Peak area estimations revealed that *Gracilariopsis lemaneiformis* had the highest total MAA content, significantly surpassing that of *Ceramium nitens* and *Pterocladiella capillacea*. This method demonstrated effectiveness and consistency in detecting MAAs [41].

In contrast, extraction with FGlcW115 yielded variable results. Palythine and asterine-330 were detected in *G. lemaneiformis* and *C. nitens,* but *P. capillacea* showed no detectable MAAs. Similarly, BGlcW115 exhibited selective detection, identifying shinorine, palythine, and Porphyra-334 in *P. capillacea* while providing questionable results for other species. Finally, UGly13 proved to be the least effective, with no detectable MAAs in *C. nitens* and only questionable results for *G. lemaneiformis* and *P. capillacea.* This suggests that NADES UGly13 may not be suitable for extracting MAAs from these macroalgae.

In the present study, MAA concentrations ranged from 0.2 to 0.5 mg/g DW in the species analysed with aqueous extractions. Future studies should focus on optimizing drying processes and collection times to enhance MAAs production. Additionally, among NADES-assisted extraction methods, both FGlcW115 and BGlcW115 effectively recovered MAAs, whereas UGly13 did not.

Due to their high solubility in organic polar solvents such as methanol, acetonitrile, and aqueous solutions, obtaining high-purity MAA extraction and selective separation is difficult [42]. Moreover, MAAs vary between different species of algae, making the extraction and separation processes even more complicated when the NADESs were tested. Furthermore, the solubility and absorbance maximum properties of MAAs depend on the polarity and pH of the extraction solvent system, making it hard to distinguish and separate them. These findings highlight the need to adjust extraction methods based on the specific characteristics of each macroalga and the MAA of interest, emphasizing the importance of optimizing techniques for cosmetic and biotechnological applications.

## 3. Materials and Methods

### 3.1. Sample Preparation

Macroalgae samples were collected in the northeast coastal areas of Colombia. The collection permits were granted by the Ministerio de Ambiente y Desarrollo Sostenible by Permission No 121 of 22 January of 2016 (modification otrosí No. 7) for research permission of marine samples. In order not to exhaust the biological resources, the most significant data were collected in Table 1. The samples were collected by SCUBA diving conducted by biologists Monica Puyana Hegedus, Brigitte Gavio, and Felipe De La Roche Zogby. A sample of each species was identified and registered for preservation and comparison, under the code shown in Table 1, in the collection of Instituto de Ciencias Naturales, Universidad Nacional de Colombia (Herbario JIWUKORI). The fresh samples were washed with sea water and dried by sun exposure for twelve hours. The cleaned material was immediately frozen and stored at −20 °C before later assays. Finally, the samples were dried one last time on the stove at 40 °C for 48 h. Each sample was ground, and the resulting material was stored in dry bottles protected from light and heat and duly labelled.

### 3.2. Chemical Screening Tests

To determine the presence of secondary metabolites such as phenolic compounds, low-polarity compounds, alkaloids, polysaccharides, and MAAs in the selected macroalgae samples, a preliminary phytochemical assay was carried out as follows:

A 500 mg sample of dry macroalgae was extracted using ultrasound (Elmasonic Easy, Singen, Germany) for 1 h, initially with 5 mL of ethyl acetate. The ethyl acetate filtrate was stored for testing the presence of low-polarity substances using thin-layer chromatography (TLC Silica gel 60 F_254_, Merck, Darmstadt, Germany), eluting with hexane/ethyl acetate 4:1, and developing with 254 nm ultraviolet light followed by 360 nm ultraviolet light (Handheld lamp, Weber Scientific, Hamilton Township, NJ, USA), and finally, with a solution of 10% (*w*/*v*) phosphomolybdic acid in absolute ethanol, followed by heating. Under these conditions, blue spots of different retention factors (Rfs) were observed, originating from compounds such as lipids, sterols, terpenoids, and other low-polarity substances soluble in ethyl acetate. The insoluble residue was extracted with ethanol under the same conditions (5 mL, 1 h, ultrasound, room temperature). The alcoholic filtrate was stored for subsequent tests. The residue was subjected to a third extraction with water (5 mL, 1 h, ultrasound, 37 kHz, 60 °C). The aqueous filtrate was collected for coloration and precipitation qualitative tests. For alkaloid detection, a new sample of 300 mg of macroalgae was taken with 5 mL of 5% HCl (1 h, ultrasound, 60 °C). The alcoholic, aqueous, and 5% HCl extracts were tested for phenolic compounds (by dropping 1% ferric chloride solution), polysaccharides (precipitation with excess ethanol), and alkaloids (precipitation with Wagner’s reagent). For the initial detection of MAAs, a short and fast test was developed: 1 g of macroalgae samples was extracted with 10 mL of water (1 h, ultrasound, 60 °C), and the filtrates were analysed by ultraviolet spectroscopy (Evolution 60S, Thermo Scientific, Madison, WI, USA) between 200 and 400 nm.

### 3.3. Preparation of the NADESs

After conducting the search for the different NADESs reported in the literature as effective solvents in the extraction of phenolic compounds for cosmetic applications, discarding choline chloride as a restricted substance in the preparation of cosmetic products, three mixtures were prepared with the following compositions: urea/glycerine 1:3 (UGly 13) [40], fructose/glucose/water 1:1:5 (FGlcW115), and betaine/glucose/water 1:1:5 (BGlcW115) [43]. The mixture of the components was prepared with calculated quantities to obtain 50 g of solvent. Each component was added to an Erlenmeyer in a magnetic stirrer: UGly13 was composed of 10 mL glycerine (99.5% Sigma Aldrich, St. Louis, MO, United States) and 13.2 g urea; FGlcW115 with 12.7 g fructose, 19.7 mL glycerine (99.5% Sigma Aldrich), and 12.5 mL deionized water; and BGlcW115 with 23.3 g betaine, 23.3 g glucose, and 11.7 mL deionized water. Subsequently, each mixture was heated to between 50 and 55 °C with constant magnetic agitation to form a colourless liquid (estimated time: 30–90 min approx.).

### 3.4. NADES Physicochemical Parameters

#### 3.4.1. Flow Profile Assay

The rheological behaviour of the prepared NADESs was observed by performing viscosity experiments in a rheometer (MCR92, Anton Paar, Graz, Austria) and using a cone and plate geometry (diameter 1 inch) at 25 °C. Samples were directly measured without dilution. Each NADES was individually poured into the geometry up to the mark shown for a defined volume. The computer program parameters were subsequently set to zero to illustrate the measurements as a function of increased shear rate and viscosity. The rheometric profile of viscosity versus shear rate of the three NADESs prepared was used to predict the rheological behaviour of the liquids obtained.

#### 3.4.2. Determination of Surface Tension

Surface tension measurements were conducted using the Du Noüy ring method at a temperature of approx. 23 °C using a Krüss K8 tensiometer (KRÜSS GmbH, Hamburg, Germany; Iridium–Platinum ring). The ring was immersed in 5.0 mL of NADES and gradually lifted above the surface. The maximum value of the force at ring detachment determined the surface tension of the interface. Each NADES was subjected to three measurements.

#### 3.4.3. Determination of pH

The pH determination of NADESs was conducted by an electrometric method, using a Metrohm pH meter model 780, integrated with a pH electrode with Metrohm temperature sensor Pt 1000 model 6.0258.600 that supports viscous and alkaline samples. This equipment was calibrated and inspected with three pH buffer solutions: HI7004, pH 4.01; HI7007, pH: 7.01; and HI7010, pH: 10.01 at 25 °C. In each measurement, a sufficient portion of prepared NADES was extracted without any kind of pretreatment until the electrode bulb was completely covered; during measurement, the solvent was kept continuously and gently agitated and the reading recorded when a stable pH value was obtained.

#### 3.4.4. Conductivity Test

The determination of the conductivity of the NADESs was conducted with a SCHOTT^®^ instruments brand conductometer, model Lab 960, with a 4-pole graphite conductivity cell SCHOTT^®^ instruments brand LF413T (Mainz, Germany). It was calibrated and verified with certified reference material for electrolytic conductivity measurement 0.001 mol/L (nominal 0.146 mS/cm) brand Merck of the Certipur^®^ line. At each measurement, a sufficient aliquot of prepared NADES was extracted without any pretreatment until the cell was completely covered, gently stirred, and conductivity and temperature were recorded.

### 3.5. Ultrasound-Assisted Extraction of Macroalgae with NADESs

Twelve samples of marine macroalgae, mentioned in Table 1, were obtained and prepared for extraction. First, 100 mg dry weight and a grounded sample of each species was weighed separately in 15.0 mL tubes. To each sample were independently added 1.5 mL of NADES, water, and 5% HCl solution to obtain 5 extracts for the same macroalgae samples: 3 individual extracts of each alga for each NADES (FGlcW115, BGlcW115, and UGly13) and 2 extracts with each of the reference solvents (water or 5% HCl solution). Subsequently, ultrasound-assisted extraction was performed at 37 kHz for 60 min at 60 °C. Each extraction was performed in triplicate.

### 3.6. Phenolic Content Determination

The analysis extracts were from the following samples: Sargassum cf. ramifolium, Sargassum fluitans, Stypopodium zonale, Pterocladiella capillacea, Solieria filiformis, Ceramium nitens, Gracilariopsis lemaneiformis, Hypnea sp. 1, Hypnea sp. 2, Dictyota menstrualis (DP120), and Dictyota cf. pulchella (DP127 and 130), obtained in each of the prepared NADESs (FGlcW115, BGlcW115, and UGly13). A microplate reader (Synergy H1 Multi-Mode Reader, BioTek, Winooski, VT, USA) equipped with a 96-well plate was used; 10 μL of the extract was added to 30 μL of Na_2_CO_3_ 20% solution, 245 μL of deionized water, and 15 μL of Folin–Ciocalteau reagent. The reading was made at 760 nm at room temperature. For pre-scalation assays, 30 μL of the extract was evaluated and 30 μL of Na_2_CO_3_ 20% solution, 225 μL of deionized water, and 15 μL of Folin reagent were added. The reading was made at 760 nm at normal temperature and pressure. The measurements were taken in triplicate.

### 3.7. Pre-Scaling of NADESs Extraction

For the scaling tests, small sample sizes were used to decrease the resistance to mass transfer to the solvent; for this, a #20 sieve with opening of 850 μm (NTC 32 A.S.T.E 11-87) was used. For the initial tests, 1.0 g of dry and ground sample of the algae *Sargassum fluitans* was weighed, and 10 mL of BGlcW115 was added. Two different suppliers of NADES components were evaluated. Extraction was performed with constant agitation using a mechanical agitator at 60 °C (Polymax 2040, Heidolph, Schwabach, Germany). Extract samples were taken after 1, 2, 4, 24, and 30 h of extraction. Then, 1.0 g of dry and ground sample of *Sargassum fluitans* was dissolved with 10, 20, and 50 mL of NADES. Extraction was performed with a mechanical agitator at 60 °C and 60 rpm and tested for phenolic compounds after at 6, 24, 30, 48, and 54 h of extraction.

### 3.8. UV Preliminary Scanning

A UV-Vis spectroscopic analysis was performed in 96-well microplates by means of a spectral sweep of 200 to 400 nm of 10 μL of the extracts of the samples *Ceramium nitens*, *Gracilariopsis lemaneiformis*, *Hypnea* sp. 1, *Hypnea* sp. 2, *Pterocladiella capillacea*, *Sargassum* cf. *ramifolium*, *Sargassum fluitans*, *Solieria filiformis,* and *Stypopodium zonale* in all solvents of interest. Water and NADESs were used as negative controls.

### 3.9. Identification of MAAs by UHPLC-DAD Analysis

MAAs extraction was performed according to the method by Urrea-Victoria et al. [41]. Briefly, 5 mg freeze-dried weight samples were spiked with 1 mL of acidified water (0.25% formic acid and 20 mM ammonium formate) and mixed thoroughly. Then, 30 mg DW of samples were extracted with 2 mL of NADES, heated at 60 °C, and subjected to ultrasound for 2 h. All MAA extractions were executed using an ultrasonic bath (Elmasonic Easy 37 kHz, Singen, Germany) for 15 min at room temperature. After extraction, the tubes were centrifuged at 1500× *g* for 5 min, and the supernatant was filtered through 0.22 µm polyvinylidene fluoride (PVDF) hydrophilic syringe filters (Thermo Fisher Scientific Inc., USA) and transferred to vials [41].

Ultra-High Performance Liquid Chromatography with Diode Array Detector (UHPLC-DAD) analyses were conducted using a Shimadzu Nexera X2 system (Shimadzu, Japan) equipped with a photodiode array detector (SPD M240). An aliquot of 5.0 μL of each filtered extract was injected into a Synergi Hydro-RP 80 Å (4 μm, 150 × 2.0 mm) column protected by a Security Guard precolumn. The flow rate was 0.3 mL/min at 22 °C. The gradient system combined an aqueous solution of 0.25% formic acid and 20 mM ammonium formate (solvent A) and acetonitrile (solvent B). It started with 100% A (0–3 min), followed by 95% A (3–7 min), and 20% A (7–8 min). The re-equilibration duration between individual runs was 5 min. The detection wavelengths were 330 nm.

### 3.10. Statistical Analysis

All the data were expressed as mean ± standard deviation derived from triplicate extractions. Statistical analyses were performed with software GraphPad, using two-way ANOVA and mixed effect analysis test, and *p* < 0.05 was considered statistically significant.

## 4. Conclusions

Based on the results obtained in this research, brown macroalgae extracts could be obtained using NADESs consisting of betaine, glucose, and water (BGlcW115), a non-toxic and environmentally friendly solvent. This study revealed that the extraction is affected by several factors such as viscosity, solid–liquid ratio, and extraction time. Significant differences in phenolic compounds were observed between the extracts of the 12 seaweed species. Brown macroalgae exhibited the highest content of phenolic compounds, particularly *S. fluitans*, the most promising species due to its phenolic compound content and abundance in the Colombian sea compared with other *Sargassum* species. This statement does not imply that the other species lack potential for further development; instead, they need additional optimization of extraction conditions using alternative NADESs, especially for red algae. In an effort to optimize, BGlcW115 was employed to extract phenolic compounds from *S. fluitans* through mechanical agitation. This yielded varying results depending on the supplier of the raw materials used to prepare the mixture and the S/L ratio between 1:20 and 1:50. Similar yields were achieved after 30 h of extraction. Identification of MAAs can be explored for the sample of *Pterocladiella capillacea*, which may contain cyclohexenimine-type MAAs such as palythine. The variation in the references of the components of the BGluW115 solvent provided an indication of its reliability, since reproducibility of the results was obtained by performing the method on the same sample under varying operating conditions. Without further purification steps, NADES extracts are proposed to be introduced directly into cosmetic products, where they would be compatible. Knowledge of the composition and stability of bioactive compounds extracted with NADESs is essential for future studies and applications to take advantage of this promising and sustainable chemical strategy.

## Figures and Tables

**Figure 1 marinedrugs-23-00239-f001:**
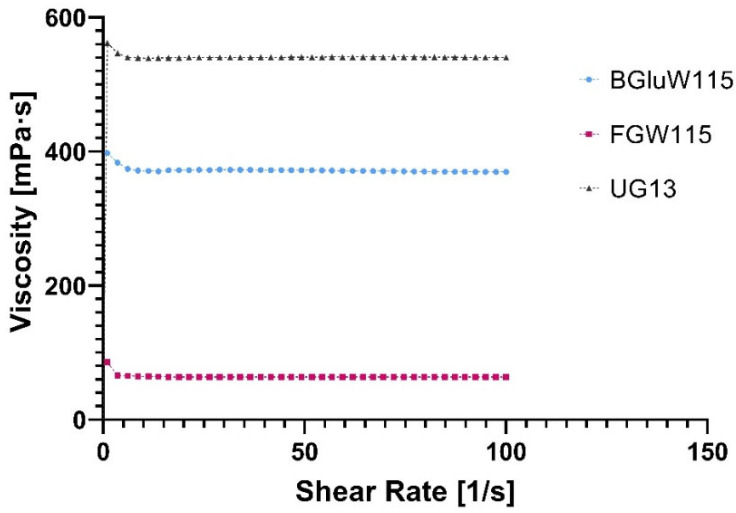
Flow profile of the prepared NADESs.

**Figure 2 marinedrugs-23-00239-f002:**
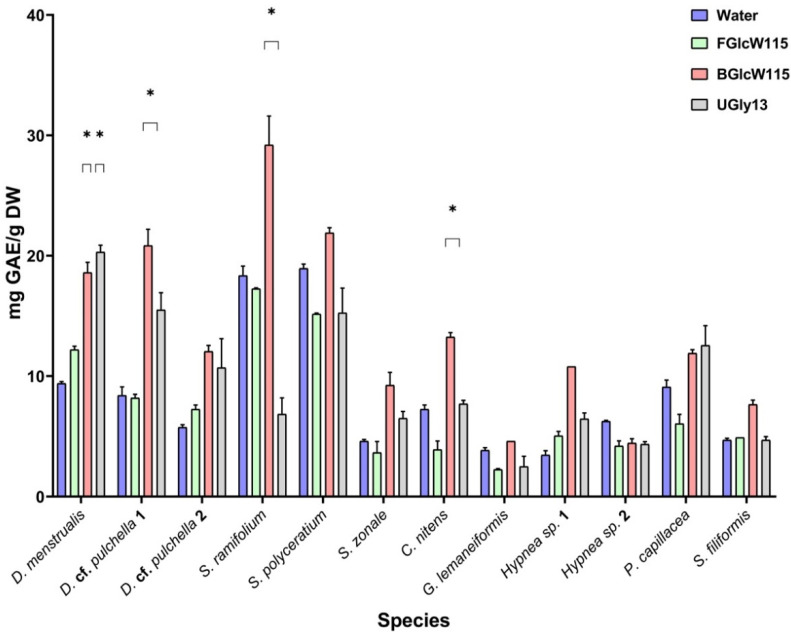
Total phenol contents of NADES extracts of different macroalgae samples. Values with * in each column are significantly different from water extraction (*p* < 0.05). Each value represents the mean ± SD of three replicates. TPC is the abbreviation of total phenolic content, expressed as mg GAE/g. GAE is gallic acid equivalent.

**Figure 3 marinedrugs-23-00239-f003:**
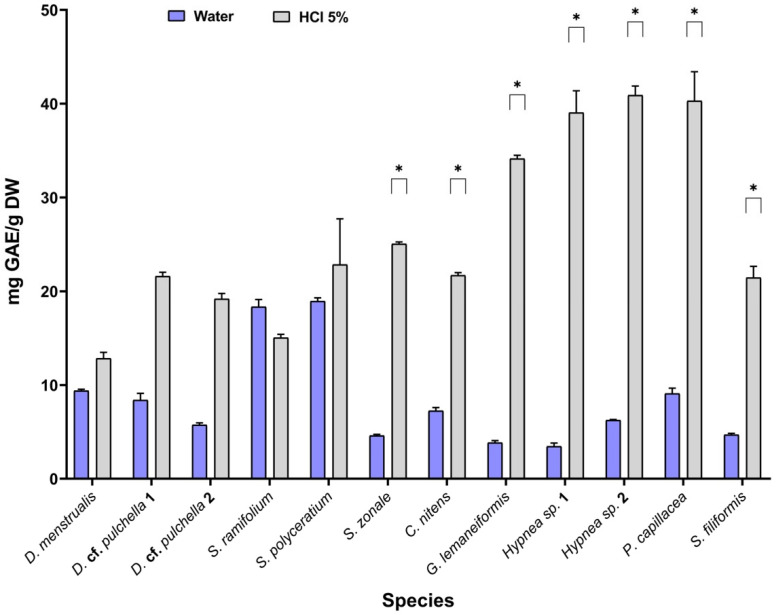
Comparative extraction of phenolic compounds between water and 5% hydrochloric acid solution. Values with * in each column are significantly different from water extraction (*p* < 0.05).

**Figure 4 marinedrugs-23-00239-f004:**
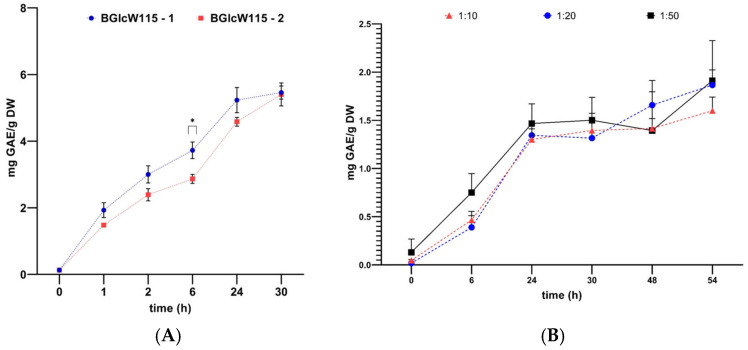
Optimization of the extraction process of TPC with BGlcW115 from *Sargassum fluitans.* (**A**) Effect of raw material source (betaine and glucose). (**B**) Effect of SLRs of 1:10, 1:20, and 1:50. Values with * in each time are significantly different between NADES (*p* < 0.05).

**Figure 5 marinedrugs-23-00239-f005:**
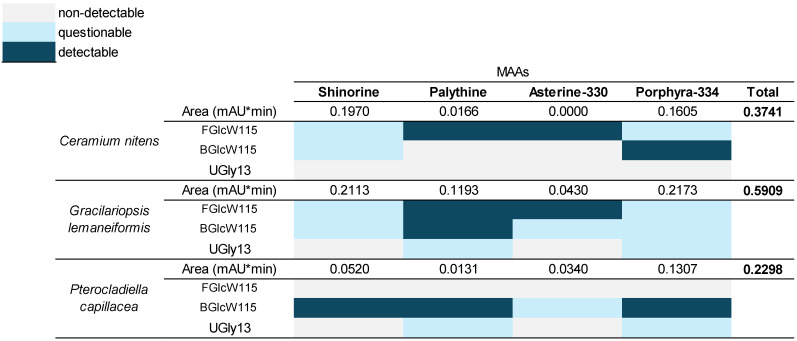
Heatmap of detection of MAAs of interest from species extracted with the mobile phase A. Dark blue: Detectable, a clear signal of the mycosporine-like amino acid in the spectrum is obtained, and the area of absorbance can be estimated. Grey: Non-detectable, no presence can be detected. Light blue: Questionable, it is possible to identify visually in the spectrum, but a sufficiently visible signal is not projected that allows estimating the area.

**Table 1 marinedrugs-23-00239-t001:** Macroalgae samples studied with codes assigned for analysis, species, collection dates, and collection sites.

Group	Species	Order	Family	Collection Date	Collection Site	Voucher Number
Brown macroalgae	*Dictyota menstrualis*	Dictyotales	Dictyoptaceae	26 September 2021	Providencia	JIW00004951
	*Dictyota* cf. *pulchella* *	Dictyotales	Dictyoptaceae	26 September 2021	Providencia	JIW00004956
	*Dictyota* cf. *pulchella* *	Dictyotales	Dictyoptaceae	26 September 2021	Providencia	JIW00004952
	*Sargassum* cf. *ramifolium*	Fucales	Sargassaceae	1 March 2021	Guajira	JIW00004940
	*Sargassum fluitans*	Fucales	Sargassaceae	1 February 2021	San Andrés	JIW00004945
	*Stypopodium zonale*	Dictyotales	Dictyotaceae	3 March 2021	Guajira	JIW00003294
Red macroalgae	*Ceramium nitens*	Ceramiales	Ceramiaceae	28 February 2021	Guajira	JIW005012
	*Gracilariopsis lemaneiformis*	Gracilariales	Gracilariaceae	1 March 2021	Guajira	JIW005011
	*Hypnea* sp. 1	Gigartinales	Cystocloniaceae	1 March 2021	Guajira	ND
	*Hypnea* sp. 2	Gigartinales	Cystocloniaceae	28 February 2021	Guajira	ND
	*Pterocladiella capillacea*	Gelidiales	Pterocladiaceae	1 March 2021	Guajira	JIW005015
	*Solieria filiformis*	Gigartinales	Solieriaceae	4 March 2021	Guajira	JIW0005029

* These macroalgae are different specimens with a pending complete taxonomical identification; ND: not determined.

**Table 2 marinedrugs-23-00239-t002:** Natural deep eutectic solvent (NADES) mixtures prepared.

Code	Components	Molar Ratio
FGlcW115	Fructose/glucose/water	1:1:5
BGlcW115	Betaine/glucose/water	1:1:5
UGly13	Urea/glycerine	1:3

**Table 3 marinedrugs-23-00239-t003:** Experimental values of viscosity (η, at 298.15 K), surface tension (γ, at 296.15 K), pH (at 295.15 K) and conductivity (κ, at 295.15 K) of the investigated NADESs.

NADES	η (mPa·s)	γ (mN·m^−1^)	pH	κ (µS·cm^−1^)
FGlcW115	63.5	67.2	7.38	2.2
BGlcW115	370.4	77.5	8.44	0.9
UGly13	540.6	66.3	9.30	1.4

**Table 4 marinedrugs-23-00239-t004:** Phenolic compounds (mg GAE/g DW).

Macroalgae Samples	Solvents			
	FGlcW115	BGlcW115	UGly13	H_2_O
*D. menstrualis*	12.2 ± 0.295	18.6 ± 0.871 *	20.3 ± 0.517 *	9.37 ± 0.168
*D.* cf. *pulchella* 1 ^a^	8.22 ± 0.306	20.9 ± 1.39 *	15.5 ± 1.38	8.44 ± 0.686
*D.* cf. *pulchella* 2 ^b^	7.26 ± 0.351	12.1 ± 0.484	10.7 ± 2.45	5.77 ± 0.240
*S.* cf. *ramifolium*	17.3 ± 0.0810	29.2 ± 2.36 *	6.86 ± 1.38	18.4 ± 0.777
*S. fluitans*	15.1 ± 0.0459	21.9 ± 0.433	15.3 ± 2.05	19.0 ± 0.359
*S. zonale*	3.63 ± 0.952	9.27 ± 1.03	6.50 ± 0.606	4.60 ± 0.184
*C. nitens*	3.91 ± 0.744	13.3 ± 0.368 *	7.69 ± 0.259	7.22 ± 0.355
*G. lemaneiformis*	2.24 ± 0.0344	2.29 ± 3.25	2.48 ± 0.875	3.83 ± 0.181
*Hypnea sp.* 1	5.08 ± 0.352	10.8 ± 0.010	6.45 ± 0.500	3.44 ± 0.342
*Hypnea sp.* 2	4.24 ± 0.436	4.47 ± 0.378	4.35 ± 0.178	6.21 ± 0.0609
*P. capillacea*	6.09 ± 0.773	11.9 ± 0.293	12.6 ± 1.60	9.08 ± 0.541
*S. filiformis*	4.94 ± 0.00740	7.66 ± 0.303	4.71 ± 0.333	4.67 ± 0.128

Means of triplicate analyses ± standard deviation. Values in the rows followed by * are significantly different than conventional solvent at *p* < 0.05 according to paired *t*-test. ^a^ corresponds to voucher number JIW00004956. ^b^ corresponds to voucher number JIW00004952.

## Data Availability

The original data presented in the study are included in the article; further information can be requested directly from the corresponding author.

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
