# Peer review of "Sustainable Extraction of Prospective Cosmetic Ingredients from Colombian Marine Macroalgae Using Natural Deep Eutectic Solvents"

_marinedrugs, 2025, doi:10.3390/md23060239_

Round 1

Reviewer 1 Report

Comments and Suggestions for Authors

I have read the manuscript sent and I have several questions and recommendations.
1. The authors do not know the theoretical basis of extraction. The theory of extraction applies to both conventional extractants and new ones, which include NADES. When using the ratio of 1.5 ml of extractant per 100 mg of crushed raw material, it is impossible to obtain an extract. Everything will be absorbed by the raw material. There is a swelling coefficient of the raw material!
And the authors extract this in a 15 ml test tube. The experiment was performed methodically incorrectly. Therefore, the data obtained by the authors cannot be interpreted in some way.
2. It is very well known that 70% acetone or alcohols or water are used to extract polyphenols. The authors chose a hydrochloric acid solution and compare it with the data for non-acidic NADES. Why???
3. Why did the authors measure the porosity of the raw material. It is known that porosity depends on many factors, including grinding. But why is this indicator needed here, in this work? To check whether it is possible to place 100 mg of raw material in a 15 ml test tube? Yes, it is.
4. Most importantly, the manuscript does not contain any information that could be useful in terms of the potential use of Colombian algae in cosmetics. It is necessary to provide a general description of the plant material, including metal contamination.

I cannot recommend this manuscript for publication in the journal.

Comments on the Quality of English Language

Some terms are used incorrectly, for example, "3 targets were prepared: the solvent target ...". What did the authors mean?

Author Response

Dear reviewer, thank you for your comments. We attached a cover letter for you explaining our revisions following your comments.

Reviewer 2 Report

Comments and Suggestions for Authors

This work studied the extraction of bioactive metabolites from twelve Colombian marine macroalgae samples. Macroalgae samples were collected from the northeast coastal areas of Colombia and extracted using ultrasound-assisted liquid extraction with different Nades and compared them with reference solvents. This is meaningful, but there are still some important issues that need to be addressed:

  1. Please add more specific experimental data to the abstract to illustrate the results.
  2. Please highlight the innovative points of this study in the introduction, such as the differences or improvements compared to existing research?
  3. Why Algae from the Northeast Coast of Colombia? Environmental influences?
  4. Please specify the number of repetitions of the experiment.
  5. The text referd to “All prepared Nades formed a eutectic mixture in different ratios, all three systems were transparent liquids”, suggesting additional NaDES images.
  6. The steps of the DPPH experiment in section 4.6 should be more detailed to facilitate experimental reproduction, e.g. reaction times need to be specific.
  7. A comparative discussion should be made by comparing the extraction rates of this study with those of previous studies.
  8. There should be spaces between numbers and units.
  9. Experimental data should have standard deviations to ensure data accuracy, please check in full.
  10. Have there been any studies on the extraction of active ingredients from marine algae by NAdes in recent years? Please add.

Overall Recommendation: Major Revision.

The amount of data in the article is insufficient and it is recommended to increase the data system.

Comments on the Quality of English Language

Needs Improvement.

Author Response

(The authors gave the same response as above.)

Reviewer 3 Report

Comments and Suggestions for Authors

Authors proposed a paper entitled: “Ecofriendly extraction of potential cosmetic ingredients from Colombian marine macroalgae, using Natural Deep Eutectic Solvents (NADES)” for the publication in Marine Drugs.

Issue 1

Abstract (Lines 22–37). The structure of the abstract is unclear and lacks a clearly stated objective. In particular, results are mentioned without briefly defining a sufficient context; the purpose of the study and its main outcome should be briefly clarified. Authors may reformulate it with a clearer flow.

Isssue 2

Introduction (Lines 41–101). Excessive repetition and awkward phrasing reduce clarity. In particular, the phrase “and the how metabolites are dissolved…” (line 51) is grammatically incorrect; the term "green" appears to be overused. Therefore, authors may remove repetitive content and clearly define the research gap.

Issue 3

Section 2.4 – Extraction with NADES (Lines 119–128). Sentence construction is confusing; unclear cause-effect relationships. In particular, "that reduction occurred without mass loss" is poorly placed and ambiguous. Authors are invited to better explain the relation between temperature, viscosity, and extraction efficiency.

Issue 4

Section 2.6 – DPPH Results (Lines 139–144). In this section, there is a lack of statistical analysis to support comparisons. Table 1 presents values without discussing their statistical significance. Authors may apply and use ANOVA or t-tests to evaluate the differences among NADES systems and include p-values.

Issue 5

Section 2.7 – Pre-scaling (Lines 147–162). In this specific part of the work, it seems that there is a not exhaustive interpretation of Figures 4–6. The effect of extraction time and solvent volume is mentioned qualitatively but lacks quantitative backing. It could be possible to include trend analysis or regression curves. Provide more detailed comments on saturation points and solvent performance.

Issue 6

Section 3.5 – Phenolic Content by Folin-Ciocalteu (Lines 283–312). Redundancy and lack of synthesis. Discussion repeats earlier results without summarizing key insights or comparing NADES vs. conventional solvents. Please, use a comparative table and focus on which solvents were most efficient and why.

Issue 7

Section 3.9 – Sample Physico-Chemical Properties (Lines 388–414). In these lines, mentions of morphology and particle size are not supported by figures or quantified observations ("data not shown"). Include microscopic images (e.g., SEM), if possible, or remove speculative content. Furthermore, if possible, clarify the impact of morphology on extraction.

Issue 8

Section 5 – Conclusions (Lines 600–638). The conclusion is only descriptive and should be more analytical. Merely summarizes results; it does not discuss implications, limitations, or future applications. Please, it would be better to highlight industrial relevance, limitations of the study, and future research directions.

Comments on the Quality of English Language

A quite good use of English, but some revisions are needed.

Author Response

(The authors gave the same response as above.)

Reviewer 4 Report

Comments and Suggestions for Authors

Dear Authors,

Please find detailed comments in the attached file below.

Kind regards

Comments on the Quality of English Language

The English could be improved to more clearly express the research.

Author Response

All suggestions were applied across the manuscript. We attached a revised manuscript with all changes highlighted and explained. Thank you for your comments.

Round 2

Reviewer 1 Report

Comments and Suggestions for Authors

I have read the manuscript after revision. I have questions that the authors did not answer.
1. The authors use extraction using 5% HCl. It is well known that acid extraction is used to extract anthocyanins. Please make a correction, namely use the correct terms. For the determination of anthocyanins, use a specific method, and not the determination by FC in eq. of gallic acid. Figure 3 must be modified and present polyphenols and anthocyanins separately. TLC method is well suited for screening anthocyanins.
2. In what conditions were the samples stored since 2021? During this time, anthocyanins, like polyphenols, are oxidized.
3. Who identified the algae samples and where are the herbarium samples stored?
4. Please provide data confirming the adequacy of the quantitative determination methods: linearity region, P2, analysis using a standard supplement.
5. Discuss the value of your algae in terms of polyphenols, anthocyanins, and other compounds found for potential use in cosmetics.
6. The list of references is not formatted correctly. It must be brought into compliance with the requirements of the MDPI. For example, reference [35] is Benoit, C., Virginie, C., & Boris, V. (2021). The use of NADES to support innovation in the cosmetic industry. In Advances in Botanical Research (Vol. 97, pp. 309-332). Academic Press.? Reference [33] is Alishlah, T., Mun’im, A., & Jufri, M. (2019). Optimization of urea-glycerin based NADES-UAE for oxyresveratrol extraction from Morus alba roots for preparation of skin whitening lotion. Journal of Young Pharmacists, 11(2), 155.? etc.

Author Response

Reviewer 1, Round 2

  1. The authors use extraction using 5% HCl. It is well known that acid extraction is used to extract anthocyanins. Please make a correction, namely use the correct terms. For the determination of anthocyanins, use a specific method, and not the determination by FC in eq. of gallic acid. Figure 3 must be modified and present polyphenols and anthocyanins separately. TLC method is well suited for screening anthocyanins.

Answer: The work was never directed to extract any class of natural compounds like anthocyanins in particular. In general, phenolic compounds were detected (ferric chloride) and quantified (R. Folin), so any result is not related to anthocyanins specifically. 5% HCl was just used as an extraction medium for preliminary phytochemical analysis for alkaloids. Precipitation tests for alkaloids require acidic aqueous media. Anthocyanins require acidic media for extraction, but preferably are extracted from fresh biological samples, because, as reviewer correctly says,  they are unstable compounds for drying and preservation during large time lapses.

Acid solutions are often added to these solvents to help stabilize the flavylium cation, which is stable in highly acidic conditions (pH ~ 3). To achieve this, the use of weak acids (e.g., formic acid, citric acid, or acetic acid) is recommended, since the use of strong concentrated acids may lead to destabilizing the anthocyanin molecule. In view of the polar structure of the anthocyanins, the addition of water to the solvent mixture can improve the extractive yield. An HCl solution at pH ~ 3 would be equivalent to a 0.00365%. Evidently, our acidic extraction was not looking for anthocyanins and just is showed as a possible underestimated effect of acidic extractions to consider in future research.

  1. In what conditions were the samples stored since 2021? During this time, anthocyanins, like polyphenols, are oxidized.

Answer: The samples were frozen and stored at − 20 â—¦C to avoid decomposition by oxidation. Assays were carried out about six months later (at 2022).

In lines 462-463 was included: “The cleaned material was immediately frozen and stored at − 20 â—¦C before latter assays.”

  1. Who identified the algae samples and where are the herbarium samples stored?

Answer: The biologists Brigitte Gavio, Felipe De La Roche Zogby and Monica Puyana Hegedus participated in collecting samples and Brigitte Gavio carried out the identification of species. The s specimens are stored in the collection of Instituto de Ciencias Naturales, Universidad Nacional de Colombia (Herbario JIWUKORI).”

In lines 457-461 was included: “The samples were collected by SCUBA diving was conducted by biologists Monica Puyana Hegedus, Brigitte Gavio and Felipe De La Roche Zogby. A sample of each specie has been identified and registered for preservation and comparison under the code en the Table 1, in the collection of Instituto de Ciencias Naturales, Universidad Nacional de Colombia (Herbario JIWUKORI).”

  1. Please provide data confirming the adequacy of the quantitative determination methods: linearity region, P2, analysis using a standard supplement.

Answer: We provide an additional text at the end of these questions and answers document with supplementary information, with data confirming the adequacy of measurements, including correlation, slopes, and Y-intercepts.

In lines 648-653 was included: “Supplementary Materials: The following supporting information can be downloaded at: https://www.mdpi.com/article/doi/s1, Table S1: Results of determination of the content of phenolic compounds in extracts from algae using the NADES FGlcW115 (r2 = 0.958); Table S2: Results of determination of the content of phenolic compounds in extracts from algae using the NADES BGlcW115 (r2 = 0.9992); Table S3: Results of determination of the content of phenolic compounds in extracts from algae using the NADES UGly13 (r2 = 0.9942).”

  1. Discuss the value of your algae in terms of polyphenols, anthocyanins, and other compounds found for potential use in cosmetics.

Answer: This work focused on phenolic compounds in general. Phenolic compounds are associated to antioxidant activity, which has beneficial effects for human consumer products including cosmetics, so there is a lot of cosmetic ingredients containing natural phenolic compounds. The other group of interest was mycosporine-like amino acids. These naturally occurring substances are known as solar filters, so they are useful as cosmetic ingredients too. Another group of compounds detected during this work were polysaccharides. Many of these compounds are used as ingredients for pharmaceutical preparations and for cosmetics. Anthocyanins were not analyzed.

In lines 556-564 was included: “Phenolic compounds are one of the bioactive compounds produced in seaweeds which have gained significant attention for skin care are considered safe with negligible cytotoxicity and many beneficial effects on humans. They vary quantitatively and qualitatively for each specimen of red, brown or green seaweeds with a wide range of studies and new developments in the pharmaceutical area and in other areas, where the predominant bioactivity of all is the anti-oxidative activity. To be more ecological and intuitive to perform, with better quality, purity, and quantity of the phenolic compounds extracted, extractions methods are still needed to be developed because seaweed phenolics can be key players in the future in different areas [29].”

  1. The list of references is not formatted correctly. It must be brought into compliance with the requirements of the MDPI. For example, reference [35] is Benoit, C., Virginie, C., & Boris, V. (2021). The use of NADES to support innovation in the cosmetic industry. In Advances in Botanical Research (Vol. 97, pp. 309-332). Academic Press.? Reference [33] is Alishlah, T., Mun’im, A., & Jufri, M. (2019). Optimization of urea-glycerin based NADES-UAE for oxyresveratrol extraction from Morus alba roots for preparation of skin whitening lotion. Journal of Young Pharmacists, 11(2), 155.? etc.

Answer: Using the bibliography software package “Zotero” we made mistakes in several references which needed to be adjusted in compliance with the requirements of the MDPI. Corrected references were:

[21]         D. Skarpalezos y A. Detsi, «Deep Eutectic Solvents as Extraction Media for Valuable Flavonoids from Natural Sources», Appl. Sci., vol. 9, n.o 19, Art. n.o 19, ene. 2019, doi: 10.3390/app9194169.

[22]         B. Y. Wong, C. P. Tan, y C. W. Ho, «Effect of solid-to-solvent ratio on phenolic content and antioxidant capacities of “Dukung Anak” (Phyllanthus niruri)», Int. Food Res. J., vol. 20, n.o 1, pp. 325-330, 2013.

[28]         Y. Yan, Pico ,Joana, Sun ,Bohan, Pratap-Singh ,Anubhav, Gerbrandt ,Eric, y S. and Diego Castellarin, «Phenolic profiles and their responses to pre- and post-harvest factors in small fruits: a review», Crit. Rev. Food Sci. Nutr., vol. 63, n.o 19, pp. 3574-3601, jul. 2023, doi: 10.1080/10408398.2021.1990849.

[29]         J. Cotas et al., «Seaweed Phenolics: From Extraction to Applications», Mar. Drugs, vol. 18, n.o 8, Art. n.o 8, ago. 2020, doi: 10.3390/md18080384.

[30]         M. Martínez-Sanz, L. G. Gómez-Mascaraque, A. R. Ballester, A. Martínez-Abad, A. Brodkorb, y A. López-Rubio, «Production of unpurified agar-based extracts from red seaweed Gelidium sesquipedale by means of simplified extraction protocols», Algal Res., vol. 38, p. 101420, mar. 2019, doi: 10.1016/j.algal.2019.101420.

[33]         T. Alishlah, A. Mun’im, y M. Jufri, «Optimization of Urea-Glycerin Based NADES-UAE for Oxyresveratrol Extraction from Morus alba Roots for Preparation of Skin Whitening Lotion – Journal of Young Pharmacists», J. Young Pharm., vol. 11, n.o 2, pp. 155-160, 2019, doi: 10.5530/jyp.2019.11.33.

[35]         C. Benoit, C. Virginie, y V. Boris, «Chapter Twelve - The use of NADES to support innovation in the cosmetic industry», en Advances in Botanical Research, vol. 97, R. Verpoorte, G.-J. Witkamp, y Y. H. Choi, Eds., en Eutectic Solvents and Stress in Plants, vol. 97. , Academic Press, 2021, pp. 309-332. doi: 10.1016/bs.abr.2020.09.009.

Reviewer 2 Report

Comments and Suggestions for Authors

I think all issues were addressed.

Author Response

Thank you for your valuable comments 

Reviewer 3 Report

Comments and Suggestions for Authors

Authors responded to my issues point by point in a complete and clear manner.  The paper has been substancially revised and can now be published.

Author Response

Thank you for your valuable comments.

Reviewer 4 Report

Comments and Suggestions for Authors

Dear Authors,

The chosen extraction method, combined with the use of NADES, represents a timely and relevant topic. Additionally, the inclusion of 12 macroalgae species adds significance to the study. However, due to the large number of comments (over 100), the manuscript in its current form does not meet the necessary standards for further consideration. A detailed review is provided in the attached file.

Kind regards

Comments on the Quality of English Language

The English could be improved to more clearly express the research.

Author Response

Dear reviewer, the attached pdf document contains point by point the corrections done to answer your comments. The new manuscript contains highlighted the respective corrections. Best regards,

Round 3

Reviewer 1 Report

Comments and Suggestions for Authors

The authors have made the necessary corrections. The manuscript can be accepted for publication.

Author Response

Thanks for your valuable comments

Reviewer 4 Report

Comments and Suggestions for Authors

Dear Authors,

The manuscript has been structured and written more clearly, which has contributed to a significant improvement in its quality. The data from the manuscript have also proven that that NADES can be an interesting alternative for the more efficient extraction of cosmetically valuable compounds.

Kind regards